# *Tlr2* Gene Deletion Delays Retinal Degeneration in Two Genetically Distinct Mouse Models of Retinitis Pigmentosa

**DOI:** 10.3390/ijms22157815

**Published:** 2021-07-22

**Authors:** Alonso Sánchez-Cruz, Andrea C. Méndez, Ignacio Lizasoain, Pedro de la Villa, Enrique J. de la Rosa, Catalina Hernández-Sánchez

**Affiliations:** 1Department of Molecular Biomedicine, Centro de Investigaciones Biológicas-Margarita Salas (CSIC), 28040 Madrid, Spain; asanchezcruz@cib.csic.es (A.S.-C.); ejdelarosa@cib.csic.es (E.J.d.l.R.); 2Neurovascular Research Unit, Department of Pharmacology and Toxicology, Facultad de Medicina, Universidad Complutense de Madrid, 28040 Madrid, Spain; ignacio.lizasoain@med.ucm.es; 3Centro de Biología Molecular Severo Ochoa (CSIC-UAM), 28049 Madrid, Spain; acanto@cbm.csic.es; 4Instituto de Investigación Hospital 12 de Octubre (imas12), 28040 Madrid, Spain; 5Department of System Biology, Facultad de Medicina, Universidad de Alcalá, 28805 Alcalá de Henares, Spain; pedro.villa@uah.es; 6Instituto Ramón y Cajal de Investigación Sanitaria (ISCIII), 28034 Madrid, Spain; 7Centro de Investigación Biomédica en Red de Diabetes y Enfermedades Metabólicas Asociadas (CIBERDEM-ISCIII), 28034 Madrid, Spain

**Keywords:** retinitis pigmentosa, retina, TLR, TLR2, microglia, innate immunity, neurodegeneration, *rd10*, *P23H*

## Abstract

Although considered a rare retinal dystrophy, retinitis pigmentosa (RP) is the primary cause of hereditary blindness. Given its diverse genetic etiology (>3000 mutations in >60 genes), there is an urgent need for novel treatments that target common features of the disease. TLR2 is a key activator of innate immune response. To examine its role in RP progression we characterized the expression profile of *Tlr2* and its adaptor molecules and the consequences of *Tlr2* deletion in two genetically distinct models of RP: *Pde6b^rd10/rd10^* (*rd10*) and *Rho^P23H/+^* (*P23H/+*) mice. In both models, expression levels of *Tlr2* and its adaptor molecules increased in parallel with those of the proinflammatory cytokine *Il1b*. In *rd10* mice, deletion of a single *Tlr2* allele had no effect on visual function, as evaluated by electroretinography. However, in both RP models, complete elimination of *Tlr2* attenuated the loss of visual function and mitigated the loss of photoreceptor cell numbers. In *Tlr2* null *rd10* mice, we observed decreases in the total number of microglial cells, assessed by flow cytometry, and in the number of microglia infiltrating the photoreceptor layers. Together, these results point to TLR2 as a mutation-independent therapeutic target for RP.

## 1. Introduction

Retinitis pigmentosa (RP) is a group of retinal genetic dystrophies responsible for the most prevalent forms of hereditary blindness. Despite affecting around 2 million patients worldwide, it is nonetheless considered a rare disease [1]. RP encompasses a range of genetically heterogeneous disorders caused by more than 3000 different mutations in over 60 genes (https://sph.uth.edu/retnet/sum-dis.htm (accessed on 6 June 2021)). There is no effective treatment to date, and although gene therapy is a promising therapeutic strategy, its applicability is limited by the genetic diversity of RP and the low prevalence of individual RP-causing mutations. Despite its genetic heterogeneity, traits common to all forms of RP include primary dysfunction and death of photoreceptors, which in turn triggers a rapid response in the microglial and macroglial cells of the retina, aimed at restoring tissue homeostasis. However, the persistence of the genetic insult leads to non-resolving and overwhelming gliosis and sterile inflammation that further contributes to disease progression [2,3]. It has been proposed that modulation of the glial response could have a disease-modifying effect [3,4]. Moreover, the targeting of common alterations could constitute a feasible strategy for treating the wide variety of RP types, regardless of the causative mutation [5,6].

Pattern recognition receptors (PRRs) are key elements of the innate immune response that engage exogenous pathogen-associated molecular patterns (PAMPs), as well as endogenous damage-associated molecular patterns (DAMPs) [7]. Activation of PRRs in tissue surveillance cells induces an inflammatory response, leading to reactive gliosis, secretion of cytokines and chemokines, and production of proinflammatory molecules [8,9]. Toll-like receptors (TLRs) are part of the PRR superfamily and in the central nervous system (CNS) are predominantly expressed by microglia [8,10,11]. A growing body of evidence implicates TLRs in neurodegenerative disease, despite some contradictory findings, which were likely conditioned by the specific TLR studied and the disease stage and etiology [8,9]. While the role of TLRs in host defense against infections of the eye surface is well established [12], little is known about their role in retinal dystrophies, particularly RP.

In the present study, we investigated the role of TLR2 in RP progression in two mouse models, the *rd10* and *P23H/+*, both of which carry mutations that cause RP in humans and recapitulate the clinical hallmarks of RP. The *rd10* (*Pde6b^rd10/rd10^*) mouse carries a recessive homozygous spontaneous missense mutation in the rod-specific phosphodiesterase 6b gene [13]. The *P23H/+* (*Rho^P23H/+^*) mouse was generated by a knock-in strategy and carries a dominant mutation, consisting of substitution of the proline at position 23 of rhodopsin gene with a histidine [14]. By using two genetically independent RP models with distinct patterns of inheritance and disease course, we sought to identify potential therapeutic targets common to a variety of RP patients.

We show that the expression of *Tlr2* and its adaptor molecules is increased in both *rd10* and *P23H/+* retinas with respect to age-matched wild type (WT) retinas. Moreover, we demonstrate that complete *Tlr2* deficiency resulted in better preservation of retinal structure and function in both RP models. In the *rd10* model, these beneficial effects correlated with a decrease in the total number of microglial cells and in the number of microglia infiltrating the photoreceptor layers. These findings point to TLR2 as a potential pharmacological target for the treatment of RP.

## 2. Results

### 2.1. Tlr2 and TLR-Adaptor Gene Expression during RP-Associated Retinal Degeneration

To examine the potential link between retinal degeneration and the innate immune response, we first analyzed expression of *Tlr2* and TLR-adaptor genes in two genetically unrelated RP mouse models, the *rd10* (*Pde6b^rd10/rd10^*) mouse and the *P23H/+* (*Rho^P23H/+^*) mouse. In addition to carrying two distinct, unrelated causative mutations, the two models differ considerably in terms of the temporal progression of the disease; in the *rd10* mouse most rods are lost within 4 weeks of birth, while in the *P23H/+* mouse rod loss occurs over several months. Reverse transcription-quantitative PCR (RT-qPCR) analysis of WT and *rd10* retinas was performed between postnatal day (P) 16 (before the appearance of evident morphological signs of retinal degeneration) and P21 (at which stage the degenerative process has become clearly established) [15,16]. This revealed a dramatic increase (over 30-fold) in *Tlr2* expression in *rd10* versus WT retinas at P21, an effect that was accompanied by a parallel increase (over 60-fold) in the expression of the proinflammatory cytokine *Il1b.* Expression levels of the adaptor genes *Myd88* and *Tirap* were over fourfold higher in *rd10* than WT retinas at P21 (Figure 1A).

Gene expression analysis of WT and *P23H/+* retinas between 3 months (early stages of retinal degeneration) and 6 months (at which point most photoreceptor cells have been lost) showed that expression of *Tlr2* and of the adaptor genes *Myd88* and *Tirap* was increased in the *P23H/+* versus WT retinas, particularly at 4 months (Figure 1B). These increases, as well as the increase in *Il1β* expression, were consistent but of a lesser magnitude than those observed in *rd10* retinas, likely reflecting the slower pace of retinal degeneration of the *P23H/+* model [14].

### 2.2. Effect of Tlr2 Deletion on Vision Loss

Given the correlation between expression levels of the innate immune response-related genes studied and both the inflammatory response and the degenerative process, we next investigated whether TLR2 was involved in the progression of RP-associated vision loss. To this end, we crossed both *rd10* and *P23H/+* mice with *Tlr2*-knockout mice in order to generate RP models with *Tlr2* allele deletions [17]. First, we investigated whether *Tlr2* hemizygosity in *rd10* mice had any impact on visual function, as assessed by electroretinography (ERG) in dark- and light-adapted conditions. We observed similar amplitudes of b-mixed (rod and cone response) and b-photopic (cone response) waves in *rd10:Tlr2^+/+^* and *rd10:Tlr2^+/−^* mice (Appendix A), indicating that deletion of a single *Tlr2* allele did not significantly affect *rd10* vision loss. However, in mice lacking both *Tlr2* alleles (*rd10:Tlr2^−/−^* mice), rod and cone (b-mixed), as well as cone (b-photopic), wave amplitudes were significantly higher than in *rd10:Tlr2^+/−^* mice (Figure 2), indicating that complete elimination of TLR2 attenuated the loss of visual function.

To determine whether the beneficial effect of TLR2 elimination on vision loss was mutation-dependent, we performed a similar study using the *P23H/+* mouse model. As observed in *rd10* mice, b-mixed and b-photopic wave amplitudes were significantly higher in *P23H/+*:*Tlr2*^−/−^ versus *P23H/+*:*Tlr2*^+/−^ mice (Figure 3). These results suggest that the beneficial effects of *Tlr2* deletion are not mutation-specific, but rather reflect a mutation-independent phenotype, supporting a role of the innate immune response in the clinical course of RP.

### 2.3. Effect of Tlr2 Deletion on Photoreceptor Preservation

Next, we investigated whether the functional effects of *Tlr2* deletion described above correlated with the maintenance of retinal structure, particularly that of the outer nuclear layer (ONL) where photoreceptors are located. After the ERG studies, histological evaluations of three retinal regions were performed: central, medial, and peripheral relative to the optic nerve. In the peripheral retina of *rd10:Tlr2*^−/−^ mice the ONL was thicker and the number of photoreceptor rows higher compared with their *rd10:Tlr2**^+/−^* counterparts (Figure 4).

The number of photoreceptor rows in the medial retina was significantly higher in *rd10:Tlr2*^−/−^ versus *rd10:Tlr2**^+/−^* retinas, although no differences were observed between genotypes in the ONL/INL thickness ratio in the medial or central areas of the retina.

The same histological analysis was performed in the *P23H/+* model. In the central retina, the relative thickness of the ONL and the number of photoreceptor rows were significantly higher in *P23H/+:Tlr2*^−/−^ versus *P23H/+:Tlr2*^+/−^ mice. In the medial and peripheral retina, a similar trend towards increases in ONL thickness and the number of photoreceptor rows was evident in *P23H/+:Tlr2*^−/−^ versus *P23H/+:Tlr2*^+/−^ mice, although these effects did not reach statistical significance (medial and peripheral ONL/INL thickness ratio, *p* = 0.08 and 0.12, respectively; number of photoreceptor rows in the medial and peripheral retina, *p* = 0.07 and 0.24, respectively; Figure 5).

Taken together, these results demonstrate a mutation-independent neuroprotective effect of *Tlr2* deletion, which preserved both photoreceptor cells and visual function in the two mouse models studied.

### 2.4. Effect of Tlr2 Deletion on the Myeloid Cell Population

As TLR2 is mainly expressed by retinal microglial cells [18,19] and microglia are the primary reactive cells to retinal damage [20,21], we next investigated whether *Tlr2* deletion affected the microglial response to RP. We used flow cytometry to analyze the retinal myeloid population in the *rd10* model, in which we had previously characterized the temporal pattern of microglial activation [22]. Retinas from *rd10:Tlr2*^+/−^ and *rd10:Tlr2*^−/−^ mice were analyzed by flow cytometry, applying the gating strategy described by O’Koren et al. [23] (Appendix A). After selecting all CD11b^+^ myeloid cells, macrophages were defined as those with high levels of F4/80 expression. Cells with an intermediate/low F4/80 expression were classified, according to CD45 expression levels, as microglia (CD45^low^) or monocytes (CD45^high^). Evaluation of Ly6C, MHC-II (I-A/I-E), and CD11c expression confirmed the identity of each myeloid cell subpopulation (Appendix A), as described previously [23]. Analysis of the three retinal myeloid cell subpopulations revealed a significant reduction in the number of microglial cells in *rd10:Tlr2^−/−^* versus *rd10:Tlr2*^+/*−*^ retinas (Figure 6). No differences were observed between genotypes in the number of macrophages or monocytes.

In agreement with the reduced number of microglial cells, RT-qPCR analyses revealed lower expression of myeloid cell genes in *rd10:Tlr2^−/−^* retinas (Figure 7A). Expression of the microglia-specific genes *P2ry12* and *Tmem119*, and that of other general myeloid genes (*Iba1*, *Cd68*, *Trem2*, and *Cx3cr1*), was decreased by about 20% in *rd10:Tlr2^−/−^* retinas with respect to their *rd10:Tlr2^+/−^* counterparts. RP-induced activation of microglia resulted in their proliferation and migration to the photoreceptor layers [20,21,24], where they phagocytosed predominantly dead, but also live, photoreceptors [25]. We scored the number of microglia that infiltrated the ONL and the outer photoreceptor segments. To this end, we used *Cx3Cr1*^GFP/+^ mice, which express GFP in myeloid cells, allowing for accurate identification of myeloid cells in wholemount retinas. The number of GFP^+^ cells infiltrating the ONL was lower in *rd10:Tlr2^−/−^:Cx3Cr1*^GFP/+^ than *rd10:Tlr2^+/−^:Cx3Cr1*^GFP/+^ retinas (Figure 7B and C). While a trend towards a similar effect was observed for GFP^+^ cells in outer segments (OS), this effect did not reach statistical significance (*p = 0*.10). A similar result was observed when the analysis was carried out in retinal sections (Appendix A). The number of CD11b^+^ cells infiltrating the ONL was decreased in *rd10*:*Tlr2*^−/−^ with respect to *rd10*:*Tlr2*^+/−^ retinas, and a similar trend was observed in the infiltration of microglia to the OS (although it did not reach statistical significance, *p* = 0.13).

Taken together, these results show that *Tlr2* deletion attenuates photoreceptor cell loss and vision impairment in distinct mouse models of RP. Furthermore, *Tlr2* deficiency leads to reductions in the number of reactive microglial cells and in microglial infiltration of photoreceptor layers.

## 3. Discussion

TLRs are key participants in the complex innate immune response triggered by tissue damage [7] and have been implicated in neurodegenerative disorders affecting different parts of the CNS [8,9], including the retina [4,26]. However, the specific roles of individual TLRs remain unclear. Here we have demonstrated that TLR2 contributes to the progression of retinal neurodegeneration in the context of RP.

Elevated expression of some TLRs, including TLR2, has been reported in the retinas of canine models of RP carrying different mutations [27], and in other non-genetic retinal dystrophies [28,29]. We found that increased expression of *Tlr2* and its adaptor molecules was associated with the progression of retinal degeneration and the retinal inflammatory response in two genetically unrelated mouse models of RP. Moreover, genetic deletion of *Tlr2* in both *rd10* and *P23H/+* mice resulted in a decreased rate of photoreceptor and vision loss in both models. In the *rd10* mouse, photoreceptor preservation was observed in the peripheral and medial retina, where a substantial number of rods remained at P36 [16,30,31]. Conversely, in the *P23H/+* mouse, which has a much slower degeneration rate, the protective effect of *Tlr2* deletion was more evident in the central retina, where photoreceptor loss was evident by 4 months of age. Blockade of TLR2 has also been shown to preserve photoreceptor cells in induced experimental models of photoreceptor degeneration [32]. Similarly, interference with the MyD88-mediated TLR signaling pathway decreases photoreceptor cell death and delays the loss of retinal function in mouse models of RP [33,34]. Beneficial effects of targeting TLR2 have also been described in animal models of other neurodegenerative pathologies of the CNS, including Alzheimer’s and Parkinson’s diseases [35,36]. However, Hooper et al. found that genetic deletion of *Tlr2* worsened light-induced retinal damage in an albino mouse model [19]. While the reasons for the discrepancies between our findings and others, and those of the latter study are not clear, differences in the type of insult and the genetic background could contribute to the divergent outcomes, as well as the fact Hooper et al. used non-pigmented mice. Moreover, microglia have been acknowledged with dual protective and harmful roles in retinal degeneration (reviewed by [37]). Although, surface receptors, disease state, and photoreceptor degeneration etiology seem to influence microglia response, the combination of signals that tilts microglia toward cytotoxic or neuroprotective functions is unknown [37].

Microglia are surveillance cells that proliferate and migrate to the affected region when retinal damage occurs. In addition to microglial reactivity, circulating monocytes invade the retina and differentiate into macrophages [3,21,24]. Our analysis of the retinal myeloid population showed that *Tlr2* deletion did not affect recruitment of monocytes and macrophages to the *rd10* retina, but did decrease microglial reactivity. The absence of TLR2 correlated with reductions in the total number of retinal microglia and in the number of microglia infiltrating the photoreceptor layers. *Tlr2* deficiency has been shown to decrease microglial proliferation in response to brain injury [38,39]. While microglia are necessary to phagocytose dead photoreceptors and cell debris, over-reactive microglia also engulf live photoreceptors [25]. Thus, inhibition of microglial phagocytosis and decreased recruitment of phagocytes into photoreceptor layers can slow photoreceptor demise and preserve visual function [25,40,41,42]. Moreover, in *rd10* mice, depletion of microglia with clodronate liposomes attenuates photoreceptor cell death [43]. Therefore, the photoreceptor preservation reported here in the absence of TLR2 could be mediated, at least in part, by modulation of microglial reactivity.

The DAMPs that trigger TLR2 activation in the context of RP remain to be established. The oxidized metabolites generated by oxidative stress have been proposed as mediators of TLR2 activation in an experimental model of photoreceptor degeneration [32]. Extracellular release of the nuclear chaperone high mobility group box-1 (HMGB1), a recognized ligand of TLR2 among others [44,45], has also been implicated in various retinal dystrophies [4,46]. Moreover, elevated HMGB1 levels have been detected in the vitreous of RP patients [47]. Future studies should seek to assess the therapeutic potential of modulating endogenous TLR2 ligands in RP.

Our findings underscore the importance and feasibility of targeting common pathways as a therapeutic strategy for the global treatment of the multiple varieties of RP caused by distinct mutations. We previously demonstrated that multi-faceted neuroprotective drugs can provide structural and functional benefits in several RP animal models [48,49,50,51]. Here, we show that the targeting of *Tlr2* has a disease-modifying effect in RP, regardless of the causative mutation; thus validating TLR2 as a potential therapeutic target for RP treatment. The present findings provide a better understanding of the mechanisms responsible for vision loss in RP, which is essential in order to identify new opportunities for intervention during the course of the disease.

## 4. Materials and Methods

### 4.1. Animals

*rd10* [13], *P23H* [14], *Cx3cr1^GFP/+^* [52], and WT control mice were obtained from The Jackson Laboratory (Bar Harbor, ME, USA). *Tlr2*^−/−^ mice had been previously generated as described [17]. All animals were bred on a C57BL/6J background, and were housed and handled in accordance with the ARVO statement for the Use of Animals in Ophthalmic and Vision Research and the guidelines of the European Union and the local ethics committees of the CSIC and the Comunidad de Madrid (Ref: PROEX 152/16, 30 June 2016; PROEX 238/16, 26 October 2016; PROEX 287/19, 17 February 2020). Mice were bred and housed at the CIB core facilities on a 12/12-h light/dark cycle. Light intensity was maintained at 3–5 lux.

### 4.2. RNA Isolation and Quantitative PCR

Total RNA from individual retinas was isolated using Trizol reagent (ThermoFisher Scientific, Boston, MA, USA). Before RT, potentially contaminating DNA was eliminated with DNAse I (ThermoFisher Scientific). RT was performed with 1 μg RNA and with a Superscript III Kit and random primers (all from ThermoFisher Scientific). qPCR was performed with the ABI Prism 7900HT Sequence Detection System, using TaqMan Universal PCR Master Mix, no-AmpEthrase UNG, and Taqman assays (Table 1) for detection (all from ThermoFisher). Relative changes in gene expression were calculated using the ΔCt method, normalizing to expression levels of the *Tbp* (TATA-binding protein) gene.

### 4.3. Histological and Myeloid Cell Analysis of Retinal Sections

Animals were euthanized and their eyes were enucleated and fixed for 50 min in freshly prepared 4% paraformaldehyde in Sörensen’s phosphate buffer (SPB) (0.1 M, pH 7.4) and then cryoprotected by incubation in increasing concentrations of sucrose (final concentration, 50% in SPB). Eyes were then embedded in Tissue-Tek OCT (Sakura Finetec, Torrance, CA, USA) and snap frozen in isopentane on dry ice. Equatorial sections (12 µm) were cut on a cryostat and mounted on Superfrost Plus slides (ThermoFisher Scientific), dried at room temperature, and stored at −20 °C until the day of the assay. Before performing further analyses, slides were dried at room temperature and rinsed in phosphate-buffered saline (PBS, pH 7.4).

For the photoreceptor row and ONL thickness evaluation, sections were stained with DAPI (4′,6-diamidino-2-phenylindole; Sigma-Aldrich Corp., St. Louis, MO, USA) for 15 min, rinsed with PBS, coverslipped with Fluoromount-G (ThermoFisher Scientific), and analyzed using a fluorescence microscope (Zeiss Axioskop, Göttingen, Germany). The number of photoreceptor rows and the ONL thickness was assessed in five sections per eye and for each section, two images of the central, medial, and peripheral retina were captured. To assess preservation of the photoreceptor layer, we compared the thickness of the ONL (which primarily contains photoreceptors) with that of the corresponding INL (which contains bipolar, horizontal, and amacrine neurons and Müller glial cell bodies). ONL thickness was normalized to that of the INL, which is not affected by degeneration at this stage [14,31], in order to correct for possible deviations in the sectioning plane. The number of photoreceptor rows was also scored. For each image, ONL and INL thickness and the number of photoreceptor rows were all measured in triplicate at random positions to obtain an average value per retinal zone per section. Measurements were acquired using the “freehand line” and “measure” tools in Fiji software.

For myeloid cell staining, sections were blocked in BGT (2.5 g/L BSA, 100 mM glycine, 0.25% (*w*/*v*) Triton X-100 in PBS) for 1 h and then incubated overnight at 4 ^°^C with anti-CD11b antibody (M1/70.15.11.5.2, Developmental Studies Hybridoma Bank, Iowa City, IA, USA) diluted in BGT. After rinsing in PBS and incubation with the anti-rat Alexa 488 secondary antibody (A-11006, ThermoFisher Scientific), sections were stained with DAPI, coverslipped with Fluoromount-G and analyzed using a laser confocal microscope (TCS SP5; Leica Microsystems, Wetzlar, Germany). The number of infiltrated microglial cells (CD11b+) was scored in the ONL and the OS, with the allocation compatible with microglia activation, of 3 retinal sections per mouse (6 images per section).

### 4.4. Microglia Quantification in Wholemount Retinas

Retinas were carefully dissected after eye fixation and incubated with DAPI for 30 min to stain nuclei. To flat-mount retinas, four opposing cuts were created in the retina, which was then mounted with Fluoromount-G between two coverslips. Wholemount retinas were analyzed using a laser confocal microscope (TCS SP8; Leica Microsystems). The number of GFP^+^ cells was scored in the ONL and OS in 4 areas (34,180 μm^2^ each) adjacent to the optic nerve. To assign each microglia cell to its respective retinal layer, optical sections of 1 μm were acquired and the z-position was visualized by DAPI nuclear staining.

### 4.5. Flow Cytometry

Both retinas from each mouse were dissected, pooled, and digested with 1.5 mg/mL collagenase A (Sigma-Aldrich) for 30 min at 37 °C in a water bath. Digestion was stopped by adding PBS−10% FBS, the retinas were mashed through a 100-µm cell strainer, and the cell suspension was then subjected to flow cytometry staining. To exclude dead cells from the analysis, samples were stained with Ghost Dye 780 (Tonbo Biosciences, San Diego, CA, USA), following the protocol recommended by the manufacturer. Next, the samples were treated with Fc block (obtained from the supernatant of the 2.4G2 hybridoma) prior to surface staining with the following antibodies: anti-CD45-eFluor450 (clone 30-F11, Biolegend, San Diego, CA, USA); anti-CD11b-PerCPCy5.5 (clone M170, Biolegend); anti-F4/80-APC (clone BM8, eBioscience, San Diego, CA, USA), anti-I-A/I-E-PE (clone M5/114.15.2, Biolegend); anti-Ly6c-biotin (clone HK1.4, Beckman Coulter, Indianapolis, IN, USA); and anti-CD11c-FITC (clone HL3BD, BD Pharmingen, San Diego, CA, USA). All samples were incubated with streptavidin-Brilliant Violet 605 (Biolegend) for anti-Ly6c-biotin detection.

Events were acquired using an FACS Canto II cell analyzer (Becton Dickinson, Franklin Lakes, NJ, USA), and all cell doublets and dead cells were excluded from the analysis. Data analysis was performed using FlowJo v10 software (Tree Star, Ashland, OR, USA).

### 4.6. Electroretinography (ERG) Recordings

Electroretinographic responses were recorded using a homemade device designed by Dr. P. de la Villa (Universidad de Alcalá, Madrid, Spain). ERG signals were amplified and band filtered between 0.3 Hz and 1000 Hz using a PowerLab T15 acquisition data card (AD Instruments Ltd., Oxfordshire, UK). Mice were maintained in the dark overnight; all recordings were performed after 12 h of absolute darkness. Animals were anesthetized in scotopic conditions with ketamine (50 mg/kg; Ketolar, Pfizer, New York, NY, USA) and medetomidine (0.3 mg/kg; Domtor, Orion Corporation, Espoo, Finland), and their pupils were dilated with a drop of tropicamide (Alcon, Fort Worth, TX, USA). Next, the ground electrode was located parallel to the tail of the animal and the reference electrode was placed in the mouth. Methocel (Colorcon, Harleysville, PA, USA) was applied to the cornea to avoid drying and the corneal electrode was placed in contact with the Methocel. The ERG recordings were first obtained in scotopic conditions with increasing light stimuli (0.1, 1, 10 and 50 cd·s/m^2^ for *rd10* mice and 0.001, 0.01, 0.1, 1, 10 and 50 cd·s/m^2^ for *P23H/+* mice). After light adaptation at 30–50 cd/m^2^ (photopic conditions), ERG response was measured at increasing light stimuli (1, 10 and 50 cd·s/m^2^ for *rd10* mice and 0.1, 1 and 10 cd·s/m^2^ for *P23H/+* mice). After ERG recording, sedation was interrupted with atipamezol (1 mg/kg; Antisedan, Orion Corporation).

All ERG wave measurements were performed by an observer blind to the experimental conditions. Wave amplitude was analyzed using Labchart 7.0 software (AD Instruments, Oxford, UK). To analyze the b-wave amplitude, we measured from the trough of the a-wave to the peak of the b-wave, according to ISCEV (International Society for Clinical Electrophysiology of Vision) standards. The peak of the b-wave was defined as the maximum value of the b-wave, ruling out the four positive peaks of the oscillatory potentials. The analysis of the ERGs was carried out by an investigator blinded to the experimental condition of each animal.

### 4.7. Statistical Analysis

Statistical analyses were performed with GraphPad Prism software 8.0 (GraphPad Software Inc., La Jolla, CA, USA). For two-group comparisons, normality was first assessed using the Kolmogorov–Smirnov normality test. For normally distributed data, a two-tailed unpaired t-test was applied. For non-normally distributed data, the Mann–Whitney *U*-test was used. Comparisons of two variables was performed using a 2way ANOVA, followed by Sidak’s multiple comparison post-hoc test in cases in which a significant interaction between both variables was found. In all cases, statistical significance was established at *p* < 0.05.

## Figures and Tables

**Figure 1 ijms-22-07815-f001:**
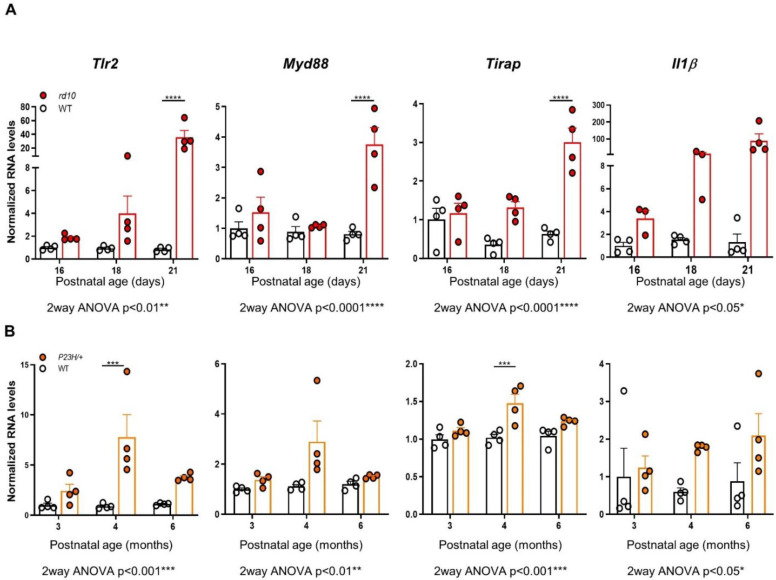
Expression of *Tlr2*, TLR adaptor genes, and *Il1b* in WT, *rd10,* and *P23H/+* retinas. Gene expression of *Tlr2,* the adaptors of the MyD88-dependent pathway *Myd88* and *Tirap,* and *Il1b* was analyzed by RT-qPCR of individual retinas from WT and *rd10* mice (**A**), and WT and *P23H/+* mice (**B**), at the indicated ages. Dots represent individual animals, and bars represent the mean (+SEM) for each group (*n* = 4 per group). Data were normalized to *Tbp* RNA and expressed relative to WT levels at 16 days (for *rd10* mice) or to WT levels at 3 months (for *P23H/+* mice). Gene expression was compared between genotypes at different time points, using 2way ANOVA, followed by Sidak’s multiple comparison test in cases in which significant interactions between age and genotype were detected. **** *p* < 0.0001; *** *p* < 0.001; ** *p* < 0.01; * *p* < 0.05.

**Figure 2 ijms-22-07815-f002:**
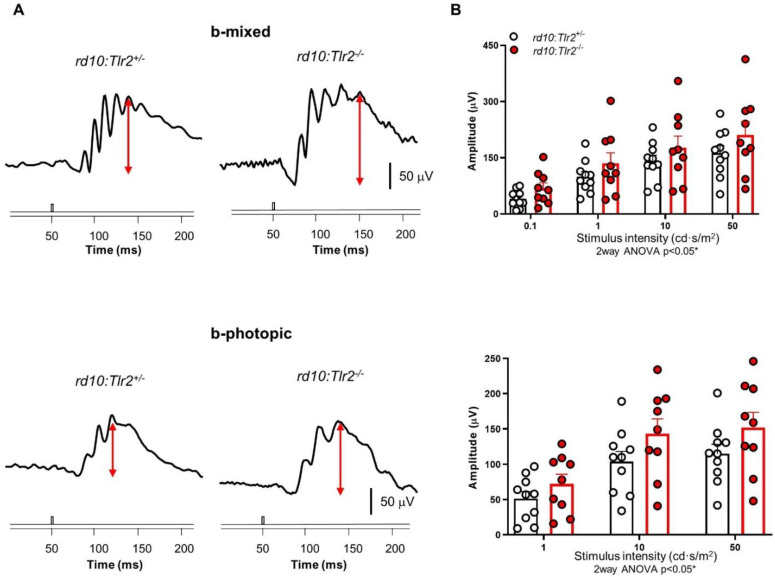
ERG response of *rd10:Tlr2*^+/−^ and *rd10:Tlr2*^−/−^ mice. (**A**) Representative ERG responses of *rd10:Tlr2*^+/−^ and *rd10:Tlr2*^−/−^ mice (P35) to a stimulus of 50 cd·s/m^2^. The value of the scale bar is indicated in the figure. (**B**) Graphs show mean ERG wave amplitudes, plotted as a function of light stimulus. Amplitudes of rod and cone mixed responses (b-mixed waves) to the indicated light intensities were recorded under scotopic conditions after overnight adaptation to darkness. Amplitudes of cone responses (b-photopic waves) to the indicated light intensities were recorded under photopic conditions after 5 min of light adaptation (30 cd·s/m^2^). Dots represent individual mice and bars represent the mean (+SEM) for each group. *n* = 9–10 animals per group. * *p* < 0.05 (2way ANOVA).

**Figure 3 ijms-22-07815-f003:**
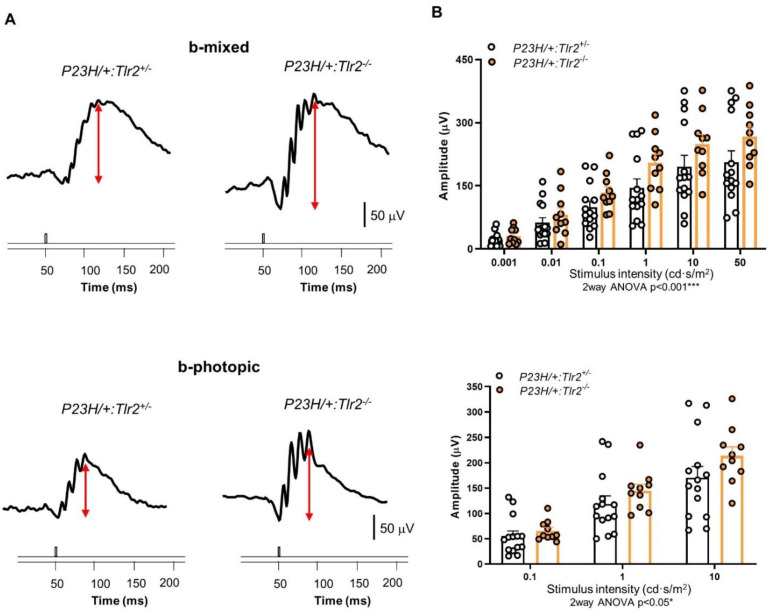
ERG response of *P23H/+:Tlr2*^+/−^ and *P23H/+*:*Tlr2*^−/−^ mice. (**A**) Representative ERG responses of 4-month-old *P23H/+:Tlr2*^+/−^ and *P23H/+:Tlr2*^−/−^ mice to light stimuli of 50 cd·s/m^2^ (b-mixed) or 10 cd·s/m^2^ (b-photopic). The value of the scale bar is indicated in the figure. (**B**) Graphs show mean ERG wave amplitudes, plotted as a function of light stimulus. Amplitudes of rod and cone mixed responses (b-mixed waves) to the indicated light intensities were recorded under scotopic conditions after overnight adaptation to darkness. Amplitudes of cone responses (b-photopic waves) to the indicated light intensities were recorded under photopic conditions after 5 min of light adaptation (30 cd·s/m^2^). Dots represent individual mice and bars represent the mean (+SEM) for each group. *n* = 10–14 animals per group. *** *p* < 0.001; * *p* < 0.05 (2way ANOVA).

**Figure 4 ijms-22-07815-f004:**
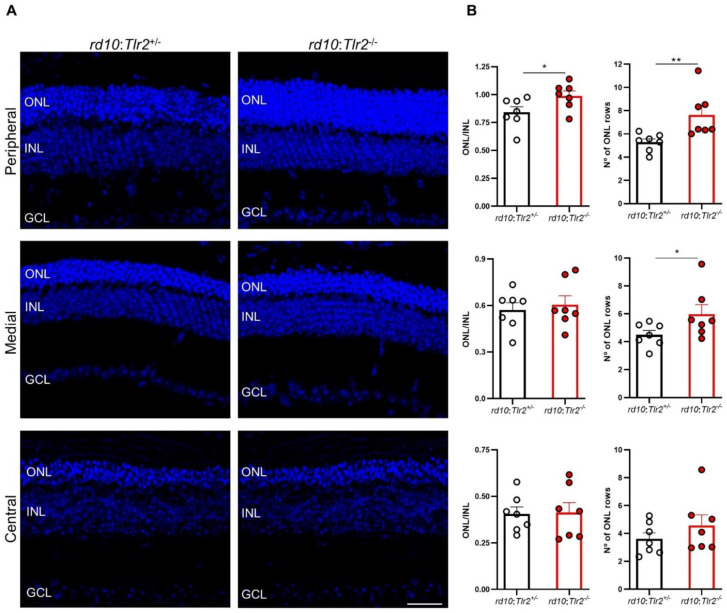
Retinal structure analysis of *rd10*:*Tlr*2^+/−^ and *rd10*:*Tlr2*^−/−^ mice. *(***A**) Representative images of peripheral, medial, and central retinal areas in cryosections from *rd10:Tlr2*^+/−^ and *rd10:Tlr2*^−/−^ mice at P36. Nuclei were stained with DAPI (blue). ONL, outer nuclear layer; INL, inner nuclear layer; GCL, ganglion cell layer. Scale bar, 31 μm. (**B**) ONL and INL thickness was measured in equatorial sections corresponding to the peripheral, medial, and central retina (see Methods section). The number of photoreceptor rows was scored in the same regions. Dots represent individual mice, and bars represent the mean (+SEM) for each group. *n* = 7 animals per group, (5 sections per retina, 2 images per section in each area, 3 measurements per image). ** *p* < 0.01; * *p* < 0.05 (unpaired *t*-test for normally distributed data (ONL/INL ratio); Mann–Whitney *U*-test for non-normally distributed data (number of ONL rows)).

**Figure 5 ijms-22-07815-f005:**
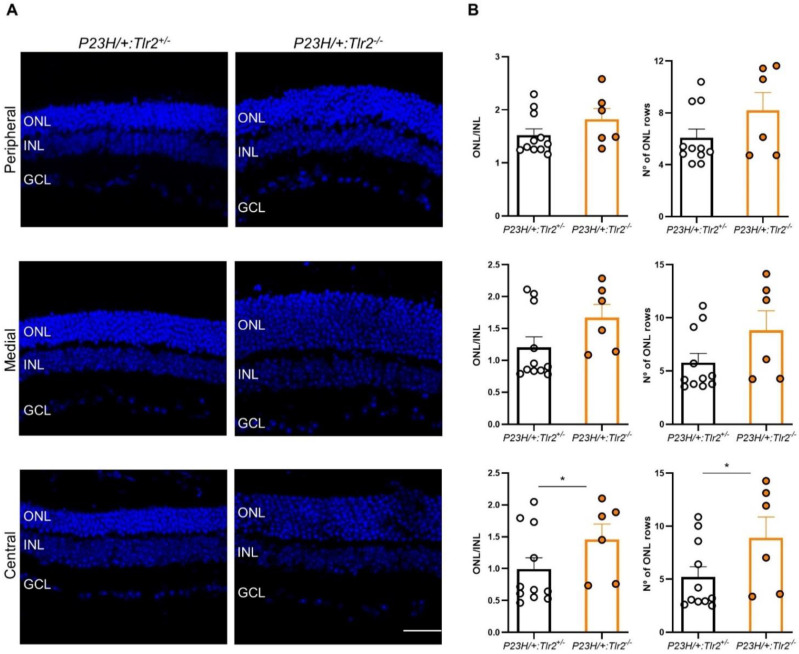
Retinal structure analysis in *P23H/+*:*Tlr2*^+/−^ and *P23H/+*:*Tlr2*^−/−^ mice. (**A**) Representative images of peripheral, medial, and central retinal areas in cryosections from 4-month-old *P23H/+*:*Tlr2*^+/−^ and *P23H/+*:*Tlr2*^−/−^ mice. Nuclei were stained with DAPI (blue). ONL, outer nuclear layer; INL, inner nuclear layer; GCL, ganglion cell layer. Scale bar: 31 μm. (**B**) ONL and INL thickness were measured in equatorial sections corresponding to the peripheral, medial, and central retina (see Methods section). The number of photoreceptor rows was also scored in these same regions. Dots represent individual mice and bars represent the mean (+SEM) for each group. *n* = 6–11 animals per group (5 sections per retina, 2 images per section in each area, 3 measurements per image). * *p* < 0.05 (Mann-Whitney *U*-test).

**Figure 6 ijms-22-07815-f006:**
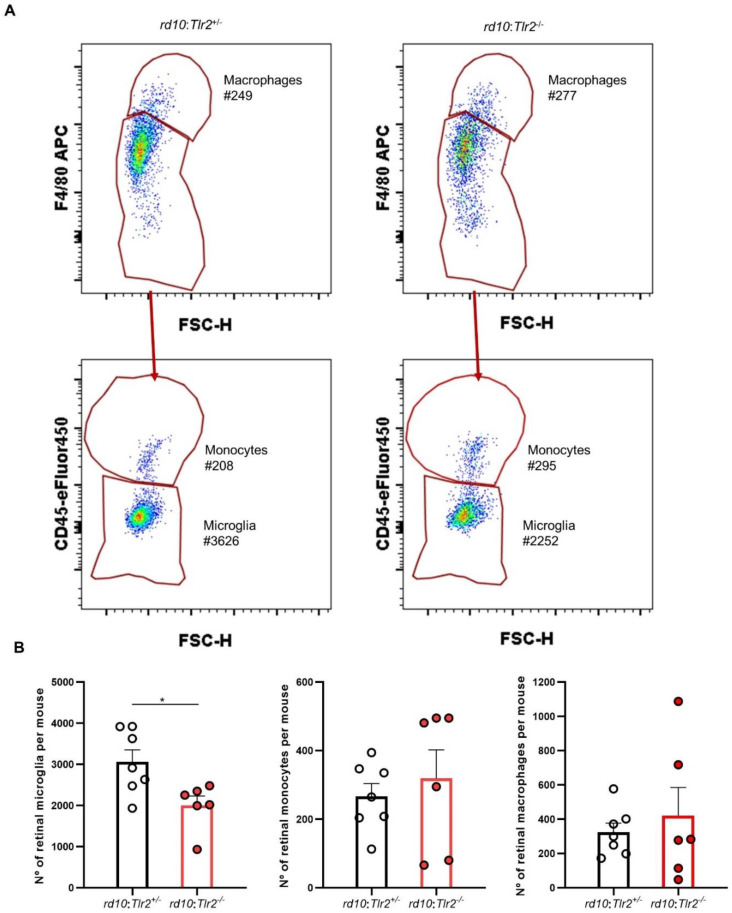
Flow cytometry analysis of myeloid cell population in *rd10:Tlr2*^+/−^ and *rd10:Tlr2*^−/−^ retinas. P25 retinas were subjected to flow cytometry analysis by surface staining with antibodies against CD11b, F4/80, and CD45, and subpopulations were defined as described in Appendix A. (**A**) Representative plots. Each gate shows the total cell count for each subpopulation. (**B**) Graphs show the number of cells in each myeloid cell subpopulation. Dots represent individual animals, while bars represent the mean (+SEM) of each group. *n* = 6–7 mice. * *p* < 0.05 (Mann-Whitney *U*-test).

**Figure 7 ijms-22-07815-f007:**
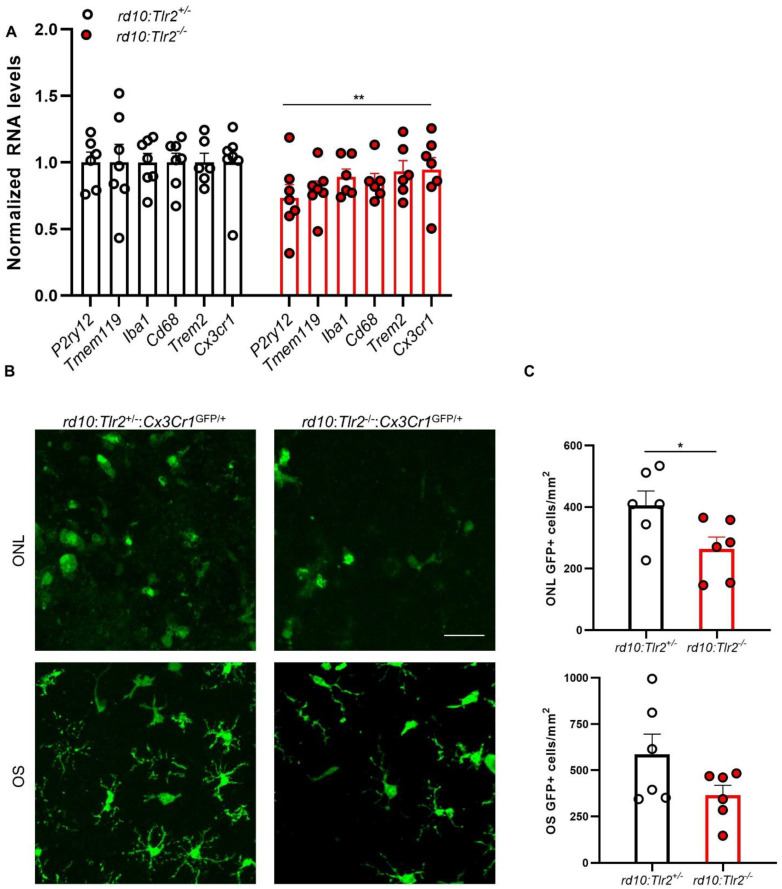
Myeloid cell analysis in *rd10:Tlr2*^+/−^ and *rd10:Tlr2*^−/−^ retinas: qPCR and histology. (**A**) RT-qPCR for myeloid cell genes in *rd10:Tlr2*^+/−^ and *rd10:Tlr2*^−/−^ retinas at P36. Transcript levels were normalized to *Tbp* RNA and expressed relative to *rd10:Tlr2*^+/−^ mice (=1). Dots represent individual animals, while bars represent the mean (+SEM) for each group. *n* = 6–7 mice. ** *p* < 0.01 (2way ANOVA). (**B**) Representative confocal optical maximal projection acquired at the indicated level in wholemount retinas from P25 *Cx3Cr1*^GFP/+^ mice. Microglial cells express GFP (green). Scale bar: 34 μm. (**C**) Quantification of the number of GFP^+^ cells in the OS and ONL of the mice shown in B. In all cases, dots correspond to individual animals and bars represent the mean (+SEM) for each group. OS, outer segment; ONL, outer nuclear layer. *n* = 6 animals. * *p* < 0.05 (unpaired Student’s *t*-test).

**Table 1 ijms-22-07815-t001:** TaqMan assays for qPCR.

Gene	Probe
*Cd68*	Mm03047343_m1
*Cx3cr1*	Mm02620111_s1
*Iba1*	Mm00479862_g1
*P2ry12*	Mm01950543_s1
*Tbp*	Mm01277042_m1
*Tmem119*	Mm00525305_m1
*Trem2*	Mm04209424_g1

## Data Availability

The datasets used and/or analyzed during the current study are available from the corresponding author on reasonable request.

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
