# Peer review of "Tlr2 Gene Deletion Delays Retinal Degeneration in Two Genetically Distinct Mouse Models of Retinitis Pigmentosa"

_ijms, 2021, doi:10.3390/ijms22157815_

Round 1
Reviewer 1 Report
The authors used double mutant mice to assess the role of TLR2 in two mouse models of retinitis pigmentosa. The results show that deleting Tlr2 in either the rd10 model or the P23H model delayed the progression of retinal degeneration, and decreased the number of retinal microglia (CD11b+ F4/80 intermediate, CD45 low) by 30%. Recently, there has been a surge of interest in studying whether the retinal mononuclear phagocytes are protective or detrimental. The current study used conclusive models and rigorous methodology. The findings will be of interest to the vision research community. The manuscript can be further improved if additional information can be provided to address the following issues. 1) As reviewed by Saban and colleagues (Ref 21), they consider microglia as mostly protective in models of retinal or RPE degeneration. This was supported by data from some studies, like Ref. 19. The discussion section of the current draft only had a very brief coverage on this important topic (line 271-274). Please elaborate on the available evidence that supports either the protective or damaging role of retinal microglia, beyond the albino mouse strain. 2) Fig. 1, some of the significant differences (Myd88 and IL1b) were not shown on the bar graphs. May have been lost during file conversion 3) ERG measurements in P23H or rd10 mice were performed at late time points. Fig. 4 and Fig. 5, the data could be strengthened if immunostaining of cone cell markers can be performed. Even though the protection against rod cell loss was marginal, the double mutant mice could have slower secondary cone loss 4) Fig. 7 showed confocal scan of microglia at different depth on retinal flat mount. It will be helpful if immunostaining of microglial marker proteins can be performed on cryosections, so that the location of microglia can be further validatedAuthor Response
Please see the attachment

Reviewer 2 Report
This article investigated the role of TLR2 gene in 2 retinal degeneration animal models. The authors found that expression level of Tlr2 and adaptor molecules increased in rd10 and P23H/+ mice. The authors also found that deletion of single Tlr2 had no effect on visual function. However, complete elimination of Tlr2 attenuated the loss of visual function in these 2 models. The authors conclude that TLR2 is a mutation independent therapeutic target for RP.
- Please provide the information on microglia activation status in these 2 animal models of TLR2 deletion (P23H/+:Tlr2-/- mice and rd10:Tlr2-/- mice). The immunofluorescence of cross section can be used to characterize the migration and infiltration of microglia.
- Page 4, figure 2 A. Please provide the detailed information on the method of determining the peak of b-wave.
Round 2
Reviewer 1 Report
As requested by both reviewers, the spatial distribution of microglia/macrophages is a key issue and has to be address with rigorous approach. The confocal images of revised Fig. 7, especially the signals from DAPI channel, did not provide enough information on the relative position of GFP-positive cells to the outer retina. Without the immunostaining data from cryosections, the manuscript cannot be recommended for further consideration.
Author Response
As requested by the reviewer we have performed microglia analysis in retinal sections. The new data are now in Figure S3 in the Appendix A section.
Reviewer 2 Report
Thank you for your reply. The migration of microglia from inner retina to outer retina is an important step during retinal degeneration. Although flat mount staining provides some advantages as addressed in the response letter, but cross section analysis will provide solid evidence on this issue. DAPI staining provides limited new information in this version.
Author Response

(The authors gave the same response as above.)

Round 3
Reviewer 1 Report
The new supplemental figure provided the requested information, although the quality of the cryosection and immunostaining was not particularly high
Author Response
Thank you
Reviewer 2 Report
Thank you for your reply.
Page 14, Line 368. Please pay attention that CD11b+ indicates activated microglia.
Author Response
Thank you for the observation. We have modified the sentence